# Position: Evaluating Generative AI Systems Is a Social Science Measurement Challenge

Hanna Wallach [1]   Meera Desai [2]   A. Feder Cooper [1]   Angelina Wang [3]   Chad Atalla [1]   Solon Barocas [1]
Su Lin Blodgett [1]   Alexandra Chouldechova [1]   Emily Corvi [1]   P. Alex Dow [1]   Jean Garcia-Gathright [1]
Alexandra Olteanu [1]   Nicholas Pangakis [1]   Stefanie Reed [1]   Emily Sheng [1]   Dan Vann [1]
Jennifer Wortman Vaughan [1]   Matthew Vogel [1]   Hannah Washington [1]   Abigail Z. Jacobs [2]

## Abstract

The measurement tasks involved in evaluating generative AI (GenAI) systems lack sufficient scientific rigor, leading to what has been described as "a tangle of sloppy tests [and] apples-to-oranges comparisons" (Roose, 2024). In this position paper, we argue that the ML community would benefit from learning from and drawing on the social sciences when developing and using measurement instruments for evaluating GenAI systems. Specifically, our position is that evaluating GenAI systems is a social science measurement challenge. We present a four-level framework, grounded in measurement theory from the social sciences, for measuring concepts related to the capabilities, behaviors, and impacts of GenAI systems. This framework has two important implications: First, it can broaden the expertise involved in evaluating GenAI systems by enabling stakeholders with different perspectives to participate in conceptual debates. Second, it brings rigor to both conceptual and operational debates by offering a set of lenses for interrogating validity.

## 1. Evaluating GenAI Systems

Evaluating a generative AI (GenAI) system[1]—i.e., making and justifying evaluative claims about that system—is critical for making decisions about whether it should be used for a particular purpose, whether it should be deployed in a particular context, or even whether it should be redesigned. The *process of evaluation*[2] necessarily requires information about the system's capabilities (like its mathematical reasoning skills), behaviors (like regurgitating pieces of its training data), and impacts (like causing its users to feel harmed). Often, this information takes the form of *measurements* on nominal, ordinal, interval, and ratio scales (Hand, 2004), where each measurement reflects the amount of some *concept of interest* exhibited by that system (related to its capabilities, behaviors, or impacts) in some *context of interest*. Such measurements are obtained via the *process of measurement*, which uses *measurement instruments*[3] (e.g., datasets, classifiers, annotation guidelines, scoring rubrics, and aggregation functions) that instantiate a particular *measurement approach* (e.g., benchmarking, automated red teaming, real-world evaluations, and user studies).

Across academia, industry, and government (e.g., National Institute for Standards and Technology, 2024; Cooper et al., 2023; Perez et al., 2022; Weidinger et al., 2023), there is an increasing awareness that the measurement tasks involved in evaluating GenAI systems are more difficult than those involved in evaluating traditional ML systems. This is because GenAI systems accept a variety of inputs, produce diverse outputs, support a wide range of use cases, and have potential impacts on people and society that range from mundane to catastrophic. As a result, concepts related to the capabilities, behaviors, and impacts of GenAI systems—the concepts to be measured when evaluating GenAI systems—are often abstract and deeply intertwined with people and society. Abstract concepts cannot be directly measured and must therefore be indirectly measured from other observable phenomena. In addition, their meanings and understandings are often contested (e.g., Mulligan et al., 2016; 2019) across—and within—use cases, cultures, and languages.

Although ML researchers and practitioners have proposed myriad measurement instruments for evaluating GenAI

---

[1]Microsoft Research  [2]University of Michigan  [3]Stanford University. Correspondence to: Hanna Wallach <wallach@microsoft.com>.

*Proceedings of the 42$^{nd}$ International Conference on Machine Learning*, Vancouver, Canada. PMLR 267, 2025. Copyright 2025 by the author(s).

[1]We use the term "GenAI system" to refer to either 1) a single GenAI model or 2) one or more integrated software components, where at least one component is an GenAI model. When we wish to refer to a single GenAI model, we use the term "GenAI model."

[2]We provide definitions for italicized terms in Appendix A.

[3]We note that measurement instruments can be qualitative or quantitative, but must collectively result in measurements.

systems, the abstract, contested nature of the concepts to be measured, coupled with the ways that measurement instruments are typically developed and used in the ML community, mean it is difficult to know precisely what these instruments are measuring and why, let alone whether they and their resulting measurements are accurate or useful—i.e., valid. In this position paper, we argue that the ML community needs to pay greater attention to the process of measurement. **We take the position that evaluating GenAI systems is a social science measurement challenge.** Specifically, the measurement tasks involved in evaluating GenAI systems—regardless of the measurement approaches and instruments used—are highly reminiscent of the measurement tasks found throughout the social sciences. Social scientists have been rigorously measuring abstract, contested concepts—ideology, democracy, media bias, framing, to name just a few—for over fifty years (e.g., Berelson, 1952; Zaller, 1992). As a result, we argue that the ML community would benefit from learning from and drawing on the social sciences when developing and using measurement instruments for evaluating GenAI systems.

We emphasize that we are not suggesting that the ML community adopt existing measurement instruments from the social sciences by transferring measurement instruments designed for humans (e.g., specific psychometric tests) to GenAI systems. Rather, we suggest standardizing the process of measurement by adopting a variant of the framework that social scientists use for measurement, shown in Figure 1. We also emphasize that although we focus on GenAI systems, this framework applies equally well to traditional ML systems. However, given the "general purpose" nature of GenAI systems, the consequences of poor evaluations are wider ranging and, in some cases, more severe. Finally, we note that we are not saying that the ways measurement instruments are typically developed and used in the ML community should be abandoned. Rather, our position is that improving the quality of GenAI evaluations means changing how the ML community conducts such evaluations—a call to action. Situating the traditional ML approach to measurement within the framework we propose can reveal gaps and limitations, helping mature current measurement practices into a rigorous science of GenAI evaluations.

**Paper roadmap:** In the next section, we discuss related work, explaining how our position systematically unifies and reinterprets work spanning multiple ML subcommunities. In Section 3, we present a four-level framework, grounded in measurement theory from the social sciences, for measuring concepts related to the capabilities, behaviors, and impacts of GenAI systems. We then describe how to use this framework in Section 4, illustrating both the core ideas and our arguments using a hypothetical running example. We also show how the framework brings clarity to existing debates about measurement in the ML community.

In Section 5, we summarize key actions for adopting the framework and discuss potential barriers to doing so. In Section 6, we present and address some views that provide alternatives to our position, before concluding in Section 7.

## 2. Related Work

**Critiques of measurement instruments:** There is a growing body of work demonstrating that the measurement instruments used in current evaluations of GenAI systems have serious limitations (e.g., Raji et al., 2021; Hutchinson et al., 2022; Rauh et al., 2024; Roose, 2024; Eriksson et al., 2025; Brandom, 2025), including conceptual confusion around precisely what is being measured, insufficient interrogation of validity, and conflicting measurements. Although much of this work has focused on benchmarking, these critiques extend to other measurement approaches, such as automated red teaming and real-world evaluations.

**Parallels to psychometric tests:** In response to these critiques, recent work has advocated for learning from and drawing on psychometrics when designing and using capability and behavior benchmarks for GenAI models (e.g., Wang et al., 2023b; Alaa et al., 2025; Salaudeen et al., 2025). These papers highlight structural similarities between benchmarks and psychological tests (e.g., items, scoring rubrics, and aggregation functions) and emphasize the importance of interrogating validity, often drawing on the foundational work of Cronbach & Meehl (1955). Although our position is similar to this recent work, it is broader in scope. Our position extends to all concepts of interest, all GenAI systems in all contexts of interest, and all measurement approaches and instruments. Because this space is both expansive and diverse, the structural similarities between benchmarks and psychological tests do not universally hold across it. We therefore argue for drawing more broadly on the social sciences, where researchers routinely measure all kinds of concepts, exhibited by diverse objects in a wide range of contexts, using a variety of measurement approaches and instruments. By taking a broader position, the framework we propose brings clarity to all measurement tasks involved in evaluating GenAI systems.

**Responsible AI:** Responsible AI (RAI) researchers and practitioners have long made similar arguments to ours, particularly in discussions related to the fairness of ML systems. Many have noted that measurements of abstract, contested concepts are hard to interpret and often reflect different meanings and understandings (e.g., Blodgett et al., 2020; Goldfarb-Tarrant et al., 2021). To address this, researchers have advocated for specifying precisely what is being measured, drawing on literature from other disciplines as appropriate (e.g., Savoldi et al., 2021; Wang et al., 2022; Katzman et al., 2023; van der Wal et al., 2024; Morehouse et al., 2025). Researchers have also emphasized the importance of interro-

gating validity (e.g., Jacobs & Wallach, 2021; Blodgett et al., 2021). However, with only a few exceptions (e.g., Liu et al., 2024; Zhao et al., 2024), these ideas have mostly been used to critique existing measurement instruments, rather than as a framework for standardizing the process of measurement.

**Benchmark consolidation:** In parallel, benchmark consolidation efforts like HELM (Liang et al., 2023), BIG-bench (Srivastava et al., 2023), DecodingTrust (Wang et al., 2023a), and TrustLLM (Huang et al., 2024) provide standardized conditions (e.g., datasets, model parameters) for evaluating GenAI models, improving both the comparability of GenAI models and the reproducibility of evaluations. However, these efforts primarily focus on benchmarking, as opposed to other measurement approaches, and are not well suited to real-world evaluations. Moreover, the benchmarks they include do not share a standardized measurement process, rarely specify precisely what they are intended to measure, and have not had their validity rigorously interrogated. Indeed, Liang et al. (2023) recognized these as weaknesses of HELM but deferred addressing them.

Collectively, the work described above, which spans multiple ML subcommunities, motivates our position. Although there are clear connections between the ideas in these papers, uptake has not been particularly fast or successful, suggesting that ML researchers and practitioners (including those working on RAI) have struggled to meaningfully incorporate these ideas into their work. We argue that by providing a cohesive, overarching way to standardize the process of measurement, our position systematically unifies and reinterprets this related work, while presenting a clear path forward. Our hope is that this will make it easier for ML researchers and practitioners to learn from and draw on these ideas when developing and using measurement instruments, in turn helping them proactively avoid a wide range of limitations (e.g., conceptual confusion around precisely what is being measured and insufficient interrogation of validity) by design.

## 3. A Measurement Framework for GenAI

When measuring abstract, contested concepts, social scientists often turn to measurement theory, which offers a structured approach to the process of measurement, as well as a set of lenses for interrogating validity (e.g., Adcock & Collier, 2001; Cronbach & Meehl, 1955; Messick, 1996). One formulation of measurement theory is the framework of Adcock & Collier (2001), a variant of which is shown in Figure 1. This variant distinguishes between four levels: the *background concept* or "broad constellation of meanings and understandings associated with [the] concept [of interest];" the *systematized concept* or "specific formulation of the concept[, which] commonly involves an

explicit definition;" the *measurement instruments*[4] used to obtain measurements of the concept; and the *measurements* themselves (Adcock & Collier, 2001). These levels are linked by four processes: *systematization*, *operationalization*, *application*, and *interrogation*. Systematization is the process of narrowing the background concept into the systematized concept; operationalization is the process of drawing on the systematized concept to develop the measurement instruments; application is the process of using the measurement instruments to obtain the measurements; and interrogation is the process of interrogating the validity of the systematized concept, the measurement instruments, and their resulting measurements. Together, these four processes comprise the process of measurement.

In Appendix C, we illustrate the widespread applicability of this framework using four very different measurement tasks: 1) measuring the prevalence of text that stereotypes social groups in the outputs of an already deployed LLM-based chatbot; 2) measuring the mathematical reasoning skills of a multimodal GenAI model; 3) measuring the extent to which a widely used LLM memorizes pieces of its training data; and 4) measuring the extent to which a company's GenAI assistant refuses to comply with harmful prompts.

### 3.1. Separating Systematization and Operationalization

As shown in Figure 1, the structured approach afforded by this framework separates systematization and operationalization, meaning that *conceptual debates* about precisely what is being measured and why—e.g., which meanings and understandings are reflected in the systematized concept? does it reflect the meanings and understandings that different stakeholders want it to reflect? if not, why?—are separated from *operational debates* about how the systematized concept is measured—e.g., do the measurement instruments yield valid measurements of the systematized concept? This structured approach differs from the way measurement is typically done in the ML community, where researchers and practitioners appear to jump from background concepts (e.g., refusal to comply with harmful prompts) to measurement instruments (e.g., a specific set of harmful prompts and a function for assessing refusal), conflating systematization and operationalization (e.g., Blili-Hamelin & Hancox-Li, 2023; Cooper et al., 2021; Jacobs & Wallach, 2021; Blodgett et al., 2020; Liu et al., 2024). However, if systematization is not treated as a separate process, resulting in a systematized concept, it is hard to know precisely what is being measured.

The separation of systematization and operationalization can

---

[4]We use the terms "measurement instruments" and "measurements" to refer to one or more measurement instruments and one or more measurements, respectively. When we wish to refer to a single measurement instrument or measurement, we use the terms "measurement instrument" and "measurement," respectively.

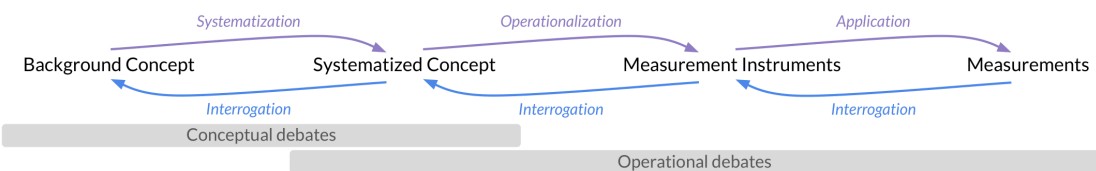

*Figure 1.* A variant of the framework of Adcock & Collier (2001). The background concept, the systematized concept, the measurement instruments, and the measurements are linked by four processes: systematization, operationalization, application, and interrogation.

also enable stakeholders with different perspectives—e.g., open-source developers, policymakers, customer, users, and members of marginalized communities, all of whom may be interested in measuring a concept for different reasons—to participate in conceptual debates and thus advocate for the inclusion of particular meanings and understandings (Abebe et al., 2020). Measuring an abstract, contested concept means making decisions about which of its meanings and understandings will be reflected in the resulting measurements. Without a systematized concept, many of these decisions are accessible only indirectly via the measurement instruments themselves, which may be hard for stakeholders other than ML researchers and practitioners to engage with. We therefore argue that adopting this framework can broaden the expertise involved in evaluating GenAI systems.

### 3.2. Emphasizing Interrogation

When measuring abstract, contested concepts, there are no directly observable, universally agreed-upon labels or scores against which to compare measurements, making interrogating validity especially difficult. [5] Different measurement theory traditions have therefore proposed different ways to interrogate validity. Like Adcock & Collier (2001), we advocate for drawing on the work of Messick (1987), who shifted away from the foundational work of Cronbach & Meehl (1955) by arguing that validity is a single, overarching concept concerning the extent of evidence supporting particular interpretations and uses of measurements. Messick also argued that the consequences of measurement instruments and their resulting measurements are fundamental to their validity. A crucial implication of this perspective is that it is not possible to interrogate validity without considering the measurement context, including the reasons for measuring the concept and how the measurements will be used. Measurement instruments and measurements that have been demonstrated to be sufficiently valid[6] in one context may not be valid in another, so validity must therefore be re-interrogated whenever measurement instruments are to be used in new contexts.

We recommend using the following set of lenses to interrogate validity, adapted from Messick (1987) by Jacobs & Wallach (2021): *face validity*, *content validity*, *convergent validity*, *discriminant validity*, *predictive validity*, *hypothesis validity*, and *consequential validity*. These lenses are primarily intended to inform operational debates, but can also shed light on conceptual debates.[7] Each lens constitutes a different source of evidence about validity. Distinguishing between the background concept and the systematized concept is crucial to obtaining meaningful evidence using the lenses. Indeed, without a systematized concept, they are not well defined. To save space, we provide an explanation for each lens in Appendix B, while also illustrating their use throughout Sections 4.1 and 4.2.

## 4. Using the Measurement Framework

In this section, we describe the systematization, operationalization, and interrogation processes in more detail. We omit a description of the application process, which consists of using the measurement instruments to obtain the measurements. To emphasize the importance of continual iteration, driven by the interrogation process, we weave our description of interrogation throughout our descriptions of systematization and operationalization. We illustrate both the core ideas and our arguments using a hypothetical running example of measuring the prevalence of text that stereotypes social groups in the outputs of an already deployed LLM-based chatbot. We also show how the framework brings clarity to existing debates about measurement in the ML community, focusing on debates about the stereotyping behaviors of LLMs and debates about the LLM-as-a-judge paradigm. In Appendix E, we additionally show how the framework brings clarity to ongoing privacy and copyright debates about the extent to which GenAI systems encode exact or near-exact copies of pieces of their training data in their parameters—a concept known as memorization.

### 4.1. Systematization

Systematization is the foundation of the process of measurement. Systematization specifies how an abstract concept

---

[5]We note that the availability of commonly used labels or scores does not mean that those labels or scores are direct observations of the concept of interest, nor that they are universally agreed upon.

[6]Determining what "sufficiently valid" means is one of the most difficult aspects of interrogating validity, as we note in Appendix B.

[7]In this regard, our views differ from those of Adcock & Collier (2001). We explain our reasoning for this in Appendix B.

is connected to observable phenomena in the real world. Specifically, systematizing a concept means taking the broad constellation of meanings and understandings associated with that concept—the background concept—and narrowing it into an explicit definition—the systematized concept—that specifies precisely what will be measured and why. We note that although the systematization process specifies how the concept of interest is connected to observable phenomena in the real world, it takes place at a theoretical level—i.e., it stops short of specifying measurement instruments.

As we explained in Section 3, the separation of systematization and operationalization can enable stakeholders with different perspectives to participate in conceptual debates. One way to do this is to directly involve them in the systematization process, giving them an opportunity to advocate for the inclusion of particular meanings and understandings.

### 4.1.1. DEFINITIONS

Suppose we wish to measure the prevalence of text that stereotypes social groups in the outputs of an already deployed LLM-based chatbot—a concept related to the chatbot's behaviors. In this example, the background concept encompasses all possible definitions of text that stereotypes social groups, making it inclusive of a broad range of meanings and understandings. First, we might more precisely define social groups—a *constituent concept* that is integral to the concept of interest. To save space, we omit a detailed description, but refer the reader to the work of Corvi et al. (2025), who define social groups as "[groups of people] who are characterized by sets of [socially salient] characteristics" and organized into social hierarchies—i.e., "systematic organizations of individuals or groups of people that differentially confer power, status, privileges, resources, and opportunities." Next, we might select a specific definition of text that stereotypes social groups, such as "[text that communicates] fixed, over-generalized belief[s] about [social groups]" (Cardwell, 1996). However, this definition still encompasses many meanings and understandings—e.g., what does "communicates fixed, over-generalized beliefs" mean?—and must be further systematized in order to specify how the concept of interest is connected to observable phenomena in the real world.

For example, we might draw on literature from social psychology, sociolinguistics, anthropology, and other relevant disciplines, finding that researchers often define the presence of such text in terms of the presence of particular linguistic patterns, such as 1) describing a social group and its characteristics in an essentializing—i.e., overgeneralized—way; 2) representing an individual as a caricature—or an essentializing representation—of a social group to which they belong; 3) prescribing essentializing ways of being for a social group and its members; and

4) proscribing essentializing ways of being for a social group and its members (Cardwell, 1996; Corvi et al., 2025).

Finally, we need to specify the relationships between the observable phenomena and the concept of interest and the relationships among the observable phenomena. Because the presence of the linguistic patterns described above define the presence of text that stereotypes social groups, the relationships between the observable phenomena and the concept of interest are definitional (as opposed to causal). All four patterns are equally important, but only one needs to be present for a piece of text to be considered stereotyping. The resulting systematized concept is depicted in Figure 2.

We note that although systematizing a concept can be challenging, it is hard to know precisely what is being measured without a systematized concept. For example, consider StereoSet (Nadeem et al., 2021) and CrowS-Pairs (Nangia et al., 2020), two widely used benchmarks for measuring the stereotyping behaviors of LLMs (e.g., assigning higher probabilities to text that stereotypes social groups). As explained by Blodgett et al. (2021), although both benchmarks provide high-level definitions of stereotyping, these definitions still encompass many meanings and understandings and do not specify how stereotyping behaviors are connected to observable phenomena in the real world. Because Nadeem et al. and Nangia et al. appear to jump from these high-level definitions to measurement instruments, it is unclear precisely what StereoSet and CrowS-Pairs are measuring. Moreover, both benchmarks' measurement instruments involve crowdworkers, who, without a systematized concept, must rely on their own understandings of these high-level definitions, which may be contradictory (e.g., whether factually true generalizations about social groups are stereotypes or not). Had Nadeem et al. and Nangia et al. further systematized their high-level definitions, they may have proactively avoided many of the limitations identified by Blodgett et al. (2021).

### 4.1.2. INTERROGATION: SYSTEMATIZATION

Continual iteration, driven by the interrogation process, is an important part of using the framework described in Section 3. Three lenses of validity—face validity, content validity, and consequential validity—are especially useful for shedding light on conceptual debates. These lenses should be used throughout systematization to interrogate the systematization process itself and the resulting systematized concept. They also provide another opportunity to involve stakeholders with different perspectives. We note that using the other lenses of validity to interrogate the measurement instruments and their resulting measurements, as described in Section 4.2.2, can also reveal systematization issues that require the systematized concept to be revised.

Returning to our running example, we might first focus on face validity. In the case of conceptual debates, face

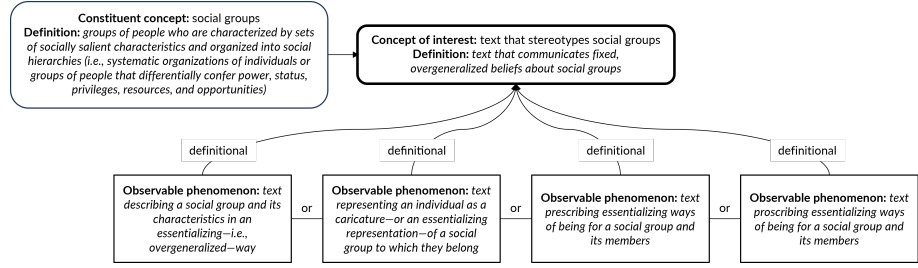

*Figure 2.* The systematized concept for our running example of measuring text that stereotypes social groups.

validity focuses on the extent to which the systematized concept looks reasonable. We might decide to seek input from members of different social groups—i.e., experiential experts—finding that they are not entirely comfortable with our systematized concept. We might therefore dig deeper, using content validity, which, in the case of conceptual debates, refers to the extent to which the systematized concept reflects the most salient aspects of the background concept. Here, we might find that members of different social groups contest our definition of text that stereotypes social groups. They might, for example, question the substantive validity of our systematized concept—i.e., whether the systematized concept fully specifies the observable phenomena that are connected to the concept of interest—perhaps by noting that the linguistic patterns do not account for differences in the acceptability of positive, negative, and neutral text that stereotypes social groups. Finally we might interrogate consequential validity, which is concerned with the consequences of measurement, including the consequences of the systematization process and the systematized concept. Here, we might find that choosing not to involve members of different social groups in the systematization process makes them feel deprioritized, disempowered, or excluded.

### 4.2. Operationalization

In contrast to the systematization process, the operationalization process takes place at an implementation level, specifying how the concept of interest is measured by drawing on the systematized concept to develop measurement instruments. Systematization and operationalization are therefore complementary: together, they ensure that the process of measurement is both theoretically and empirically grounded. We note that in some cases, there may be existing measurement instruments that can be repurposed, with or without modification; in other cases, the measurement instruments must be developed from scratch. However, in all cases, the validity of the measurement instruments and their resulting measurements must be interrogated before they are used.

#### 4.2.1. MEASUREMENT INSTRUMENTS

Having systematized the concept of interest, the first step in the operationalization process is to specify how the observable phenomena will be represented by defining a set of variables—often called *indicators*[8]—that reflect the observable phenomena. Continuing with our running example, we might define a binary-valued variable for each linguistic pattern, whose value indicates the presence or absence of that linguistic pattern in a single chatbot output.

Next, we must define how the values of the indicators should be aggregated, making sure that the aggregation functions reflect the relationships between the observable phenomena and the concept of interest and the relationships among the observable phenomena specified during the systematization process. In the case of our running example, taking the maximum of the indicators' values for a single chatbot output yields a binary value that reflects the presence or absence of any of the linguistic patterns in that output. Averaging these values over the entire population of outputs yields the prevalence of text that stereotypes social groups.

Having defined both the indicators and how their values should be aggregated, the next step is to develop the measurement instruments—i.e., the operational procedures and artifacts used to obtain the measurements. In some cases, we might wish to use a single measurement instrument to obtain and aggregate the values of the indicators; in other cases, we might wish to use multiple measurement instruments.

Returning to our running example, we might decide to develop an LLM-as-a-judge system (Zheng et al., 2023). First, we might use our systematized concept and the definitions of the indicators to develop annotation guidelines that serve as prompt instructions for a judge LLM. These annotation guidelines might include the definitions from Section 4.1.1, along with examples, counterexamples, and contextual explanations. Given a single chatbot output from a dataset that represents the entire population of chatbot outputs, the annotation guidelines would instruct the judge

---

[8]As we discuss in Appendix A, there is some inconsistency in the use of the term "indicators" in the social science literature.

LLM to generate an annotation for that output in the form of a binary vector indicating the presence or absence of each of the linguistic patterns in that output. We would then apply a per-output aggregation function to each such vector (in this case, taking the maximum of its values) before applying a population-level aggregation function to the resulting per-output values (in this case, averaging them over the dataset of chatbot outputs) to yield an estimate of the prevalence of text that stereotypes social groups in the chatbot's outputs.

There are many decisions that can influence the quality of a judge LLM's annotations, most notably the choice of model, the prompt instructions, and the configuration settings (e.g., the decoding strategy, temperature, and other parameters). But, in the case of the approach described above, the most consequential decisions are arguably those that led to the systematized concept and the definitions of the indicators and aggregation functions, all of which are necessary to produce granular, indicator-level annotations that can then be aggregated to yield the desired measurement.

In practice, ML researchers and practitioners rarely systematize their concepts of interest or provide clear definitions of indicators and aggregation functions. Instead, they typically use high-level annotation guidelines, often including illustrative few-shot examples to guide their judge LLMs' reasoning through in-context learning (Brown et al., 2020). Although this approach may end up yielding valid measurements, validity should not be assumed and must be rigorously interrogated (Pangakis & Wolken, 2025).

### 4.2.2. INTERROGATION: OPERATIONALIZATION

All seven lenses of validity can inform operational debates. These lenses should be used throughout operationalization to interrogate the operationalization process itself, the measurement instruments, and their resulting measurements. Although it is possible to develop measurement instruments without first systematizing the concept of interest, the lenses of validity are not well defined without a systematized concept, as we explained in Section 3.2. We also note that in some cases, using the lenses can also reveal systematization issues that require the systematized concept to be revised.

Continuing with our running example, we would likely begin by constructing an unannotated validation dataset of chatbot outputs to which we would then apply the measurement instruments. From there, we might first interrogate face validity by asking a colleague whether the judge LLM's annotations for 20 randomly selected outputs look reasonable to them, finding that the annotations for four outputs differ from our colleague's expectations. This does not mean that those annotations are incorrect, nor that the rest are correct, but it does merit further investigation. Whether the judge LLM's annotations are correct depends on whether they align with our systematized concept, not

our colleague's determination. For example, suppose the judge LLM determines that the text "she's amazing at math for a woman" stereotypes women, but our colleague disagrees. Who is correct? Because our systematized concept specifies essentializing descriptions as an observable phenomenon, this text does stereotype women—i.e., our colleague is incorrect and the judge LLM is correct. But there are other systematized concepts for which our colleague might be correct, such as systematized concepts where essentializing descriptions were explicitly excluded. This example highlights the importance of using lenses other than face validity to obtain less subjective evidence.

Next, we might interrogate content validity, which, in the case of operational debates, refers to the extent to which the measurement instruments align with the substance and structure of the systematized concept. By analyzing the judge LLM's annotations for our validation dataset, we might find that the judge LLM often incorrectly annotates chatbot outputs about individuals, without any reference to their social group membership, as containing text that stereotypes social groups. This constitutes evidence against the substantive validity—i.e., the extent to which the measurement instruments align with the observable phenomena specified as part of the systematized concept—of the judge LLM. Turning to structural validity—i.e., the extent to which the measurement instruments align with the relationships specified as part of the systematized concept—we might find that our per-output aggregation function did not appropriately aggregate the values of the indicators (e.g., by computing a sum rather than taking the maximum).

Interrogating convergent validity—i.e., the extent to which the measurement instruments yield measurements that are similar to measurements of the concept of interest, or other similar concepts, obtained using other, already validated measurement instruments—currently poses a challenge for the ML community because systematization and operationalization are very often conflated and few already validated measurement instruments exist. In practice, ML researchers and practitioners often compare their judge LLMs' annotations to annotations produced by human annotators, reporting inter-annotator agreement rates among the human annotators and between the human annotators and their judge LLMs (Gu et al., 2024). The core assumption underlying such comparisons is that the human annotators are already validated measurement instruments. Moreover, when human–LLM inter-annotator agreement rates are described as judge LLM "accuracies," the humans' annotations are implicitly being treated as "ground truth." This practice, although common, is problematic. Without a systematized concept there is no "ground truth"—and even with a systematized concept, unless there is evidence that the humans' annotations align with it, the human annotators cannot be viewed as already validated measurement instru-

ments (Mikhaylov et al., 2012). At a minimum, the people who systematized the concept must assess whether the humans' annotations align with the systematized concept (Halterman & Keith, 2024). Although currently uncommon in the ML community, Yu et al. (2023) took steps toward this when measuring refusal in the context of GenAI jailbreaking by directly comparing their judge LLM's annotations to their own annotations of refusal based on their systematized concept, obviating the need to validate annotations produced by human annotators. In the case of our running example, assessing whether the humans' annotations align with the systematized concept would involve analyzing the humans' annotations for an additional unannotated validation dataset. Here, we might find that some occurrences of low inter-annotator agreement are actually due to systematization issues that require us to revise our systematized concept. Having done this, we would then revise the annotation guidelines, obtain new annotations, and once again conduct a thorough analysis, continuing to iterate until the humans' annotations align with the systematized concept. Only once alignment has been reached is it possible to interrogate convergent validity by comparing the judge LLM's annotations to the humans' annotations for the original validation dataset.

Predictive validity focuses on the extent to which the measurements predict observable phenomena that are external to the concept of interest—i.e., distinct from those specified as part of the systematized concept—but known to be related to it in some way. Interrogating predictive validity often involves making predictions about real-world data. For example, we might investigate whether our measurements predict the prevalence of users' complaints about text that stereotypes social groups. If they do not, this suggests systematization issues, operationalization issues, or both. To try to rule out systematization issues, we might review the complaints to determine whether users' understandings of text that stereotypes social groups align with our systematized concept. If they do, this suggests operationalization issues.

Finally, we might interrogate consequential validity, focusing on the consequences of the judge LLM and its resulting measurements. Here, we might find that using the measurements to suppress text that stereotypes social groups may lead to the suppression of desirable chatbot outputs, such as those generated when members of different social groups ask the chatbot for advice about their lived experiences.

To save space, we omit detailed discussions of how we might interrogate discriminant validity and hypothesis validity, but provide high-level overviews in Appendix D.

Having used the lenses of validity to interrogate the operationalization process, the measurement instruments, and their resulting measurements, we can move on to the application process provided no further iteration is required.

## 5. Adopting the Measurement Framework

Below, we summarize key actions for ML researchers and practitioners who wish to adopt the framework we propose.

**Engage in conceptual debates:** As appropriate, draw on literature from relevant disciplines to systematize the concept of interest. Directly involve stakeholders with different perspectives, all of whom may be interested in measuring the concept for different reasons, in the systematization process. Use face validity, content validity, and consequential validity to shed light on conceptual debates, again seeking input from stakeholders with different perspectives.

**Engage in operational debates:** Interrogate the validity of the measurement instruments and their resulting measurements using the lenses of validity. When doing so, distinguish between the background concept and the systematized concept. As appropriate, involve stakeholders with different perspectives. Re-interrogate validity before using the measurement instruments in new measurement contexts.

**Share the systematized concept:** Share a description of the systematized concept along with the measurement instruments and their resulting measurements to make clear precisely what is being measured. This makes it easier to identify—and therefore avoid—apples-to-oranges comparisons. It also makes it easier for others to interrogate validity.

**Share evidence of validity:** Share information about the ways in which the validity of the systematized concept, the measurement instruments, and their resulting measurements were interrogated. Also share the resulting evidence for—and against—their validity. This makes it easier for others to make informed decisions about using the measurement instruments or their resulting measurements.

We note that our descriptions of the framework itself in Section 3 and its use in Section 4 reflect an ideal. Fully adopting this ideal via the key actions summarized above may be challenging for ML researchers and practitioners, especially those with limited time, budgets, or other resources. We recognize this and argue that even partial adoption will meaningfully contribute to changing how the ML community conducts GenAI evaluations. For example, situating the traditional ML approach to measurement within the framework we propose can reveal gaps and limitations.

Of the key actions summarized above, the most important are 1) separating systematization and operationalization and 2) interrogating validity. However, these actions, as well as the others, can be undertaken with varying levels of comprehensiveness depending on the resources available and the purpose for which the measurements will be used. For example, a researcher who wishes to undertake a particular measurement task to inform the next step in their project may only need to use a lightweight systematization process and

a few lenses of validity. In contrast, a team developing measurement instruments for conducting pre-deployment evaluations of a large corporation's GenAI systems likely needs to be much more comprehensive. As another example, a small university lab conducting a third-party evaluation of a newly released GenAI model may wish to be as comprehensive as possible within the bounds of their available resources.

Finally, we acknowledge that changing how the ML community conducts GenAI evaluations will be challenging. Rather than placing the entire burden on individual ML researchers and practitioners, we anticipate that uptake will be faster and more successful if organizations provide support in the form of resources and incentives, as well as working to identify and remove any other organizational barriers to adoption.

## 6. Alternative Views

In this section, we present and address some views that provide alternatives to our position, reflecting actual conversations we have had about evaluating GenAI systems.

**Current evaluations of GenAI systems may be flawed but they kind of work and everyone uses them. Do we really need something different?** As GenAI systems are deployed in more and more real-world contexts, there is an increasing awareness that evaluations that "kind of work" are no longer sufficient. Indeed, it is widely understood that current evaluations have serious limitations (e.g., Raji et al., 2021; Hutchinson et al., 2022; Rauh et al., 2024; Roose, 2024; Eriksson et al., 2025; Brandom, 2025). As Maslej et al. (2024) argued, "the lack of standardized evaluation makes it extremely challenging to systematically compare the limitations and risks of [GenAI systems]." The framework described in Section 3 is one proposal for standardizing the process of measurement, making it easier to see when and why measurements can be compared. Since there is already a desire to standardize evaluations of GenAI systems, the ML community would be well served by drawing on other disciplines as appropriate, rather than starting from scratch.

**This framework only seems necessary when ML researchers and practitioners are not already clearly stating and interrogating their assumptions.** Although ML researchers and practitioners have a variety of practices for engaging with assumptions, the framework described in Section 3 offers a structured approach to the process of measurement, including stating and interrogating assumptions, thereby unifying and reinterpreting these practices.

**GenAI systems are computational systems, not social systems, so the social sciences are not relevant to GenAI evaluations.** The "general purpose" nature of GenAI systems, combined with their increasingly wide deployment, mean that many concepts related to their capabilities, behaviors, and impacts are abstract and deeply intertwined

with people and society. We argue that these concepts more closely resemble the concepts traditionally measured by social scientists than those typically measured by ML researchers and practitioners. As a result, we believe that the ML community should adopt a variant of the framework that social scientists use for measurement. We emphasize that we do not mean to suggest that GenAI systems should be anthropomorphized, nor that measurement instruments designed for humans would be valid if applied to GenAI systems.

**Getting the ML community to adopt this framework will be a lot of work. Is the juice really worth the squeeze?** Changing the current state of GenAI evaluations will be a lot of work regardless of exactly how it is done. However, we note that the separation of systematization and operationalization parallels existing separations that have led to advancements in computer science. For example, Amdahl et al. (1964) described the separation between the logical structure and the physical realization of the IBM System/360. This separation was a pivotal innovation in computer architecture. As another example, the separation of protocol definitions and their concrete implementations at endpoints is fundamental to internet measurement (Saltzer et al., 1984). Finally, in the context of programming languages, Kowalski (1979) distinguished between the logic component and the control component of an algorithm, arguing that "computer programs would be more often correct and more easily improved and modified if their logic and control aspects were identified and separated."

## 7. Conclusion

Although GenAI systems are increasingly widely deployed, the current state of GenAI evaluations leaves much to be desired. We take the position that evaluating GenAI systems is a social science measurement challenge. We argue that the ML community would benefit from learning from and drawing on the social sciences when developing and using measurement instruments for evaluating GenAI systems. Specifically, we suggest standardizing the process of measurement by adopting a variant of the framework that social scientists use for measurement. With this goal in mind, we presented a four-level framework, grounded in measurement theory from the social sciences, for measuring concepts related to the capabilities, behaviors, and impacts of GenAI systems. We also showed how this framework brings clarity to existing debates about measurement in the ML community. Finally, we presented and addressed some views—drawn from our experiences—that provide alternatives to our position. To summarize, maturing current measurement practices into a rigorous science of GenAI evaluations will require the ML community to pay more attention to the process of measurement. We believe this would be best done by learning from and drawing on the social sciences.

## Acknowledgments

This work was supported in part by the Microsoft Research AI & Society Fellows Program. We thank Agathe Balayn, Doug Burger, Susan Dumais, Anna Kawakami, Tim Vieira, our ICML reviewers, and others for helpful suggestions. We also thank the organizers of the NeurIPS 2024 workshop on "Evaluating Evaluations: Examining Best Practices for Measuring Broader Impacts of Generative AI" for giving us an opportunity to present and refine our ideas.

## Impact Statement

By suggesting that the framework described in Section 3 can improve evaluations of GenAI systems, we do not mean to suggest that it will inevitably improve how GenAI systems are developed, deployed, used, or regulated. Indeed, the social sciences have repeatedly demonstrated that a better understanding of a problem does not automatically translate into better policies or practices. Although we believe the framework can help clear up conceptual confusion, broaden the expertise involved in evaluating GenAI systems, and yield more valid measurements, it needs to be accompanied by sustained efforts to meaningfully inject research into policymaking and practice (e.g., Cooper et al., 2024).

Because the measurement instruments typically developed and used in the ML community tend to be quantitative, we risk being misunderstood as suggesting that the framework described in Section 3 is only suitable for quantitative measurement instruments. In fact, although measurements themselves are necessarily quantitative, measurement instruments can be qualitative or quantitative, and the framework supports both. Moreover, Adcock & Collier (2001) stated that their framework, which forms the basis of ours, was intended to be a shared standard that would allow "quantitative and qualitative scholars to assess more effectively, and communicate about, issues of valid measurement."

Finally, we emphasize that adopting the framework described in Section 3 is not a panacea. Even when evaluations of GenAI systems are grounded in measurement theory, they may still fall short of what they are intended to accomplish. Indeed, the framework will often reveal such shortcomings. Rather than thinking of the framework as a solution to all the problems that beset evaluations of GenAI systems, we think of it as providing a cohesive, overarching way to standardize the process of measurement, making clear precisely what measurement instruments are and are not measuring.

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

## A. Terminology

### A.1. Evaluating GenAI Systems

**Object of interest:** The GenAI system to be evaluated.

**Concept of interest:** The concept that is to be measured. Usually related to the object of interest's capabilities (like its mathematical reasoning skills), behaviors (like regurgitating pieces of its training data), or impacts (like causing its users to feel harmed). Often abstract and therefore cannot be directly measured. May have contested meanings and understandings—across and within—use cases, cultures, and languages (Mulligan et al., 2016; 2019).

**Context of interest:** The context in which the concept of interest, as exhibited by the object of interest, is to be measured. Examples include adversarial use and real-world deployment contexts. Sometimes left unspecified.

**Measurements:** Quantities on nominal, ordinal, interval, or ratio scales. Outputs of the measurement process. Each measurement reflects the amount of the concept of interest, as exhibited by the object of interest in the context of interest.

**Measurement instruments:** The operational procedures and artifacts used to obtain the measurements. Examples include datasets, classifiers, annotation guidelines, scoring rubrics, and aggregation functions.[9] Can be qualitative or quantitative, but must collectively result in measurements.

**Measurement approach:** The approach that the measurement instruments instantiate. Consists of a strategy—i.e., a high-level protocol for obtaining the measurements—and a scope—i.e., the boundaries of what will and will not be measured. Examples include benchmarking, automated red teaming, real-world evaluations, and user studies.

**Measurement process:** The systematic process by which the measurements are obtained. Uses measurement instruments that instantiate the measurement approach.

**Evaluation process:** The broader process of making and justifying evaluative claims about the object of interest, often to inform decisions. Requires information about the object of interest, often in the form of measurements of various concepts of interest, as exhibited by that object in various contexts of interest. Example evaluative claims include: the object should or should not be used for a particular purpose, the object should or should not be deployed in particular contexts, and the object should or should not be be redesigned.

A measurement therefore reflects the amount of a concept of interest, as exhibited by an object of interest in a context of interest. Measurements are obtained via the

---

[9] We note that we take an expansive view of measurement instruments. For example, we include functions that aggregate the values of other measurement instruments to emphasize that aggregation is an integral part of the measurement process.

measurement process using measurement instruments that instantiate a measurement approach. Measurements are one form of information used to make and justify evaluative claims about an object of interest, often to inform decisions.

### A.2. A Measurement Framework for GenAI

We use slightly different terminology to that of Adcock & Collier (2001). However, the core ideas are very similar.

**Background concept:** The "broad constellation of meanings and understandings associated with [the] concept [of interest]" (Adcock & Collier, 2001). The meanings and understandings associated with the concept may be contested.

**Systematized concept:** The "specific formulation of the concept[, which] commonly involves an explicit definition" (Adcock & Collier, 2001). Very often omitted or insufficiently precisely specified in the typical ML approach to measurement. Distinguishing between the background concept and the systematized concept is crucial to obtaining meaningful evidence when interrogating validity.

**Constituent concepts**: Concepts that are integral to the concept of interest. Must be defined during systematization.

**Measurement instruments:** See above. In the context of the framework described in Section 3, the measurement instruments should operationalize the systematized concept.

**Indicators:** A set of variables that reflect observable phenomena in the real world that are connected to the concept of interest. Defined at the start of the operationalization process. The values of the indicators are obtained and aggregated using the measurement instruments. We note that there is some terminological inconsistency in the social science literature, with some researchers using the term "indicators" to refer to the observable phenomena themselves, others using it to refer to the variables that represent the observable phenomena, and others still (including Adcock & Collier (2001)) appearing to use it to refer to variables that represent the observable phenomena and the instruments for obtaining and aggregating their values.

**Measurements:** See above.

**Systematization:** The process of narrowing the background concept into the systematized concept. The foundation of the measurement process. Very often conflated with operationalization in the typical ML approach to measurement.

**Operationalization:** The process of drawing on the systematized concept to develop the measurement instruments.

**Application:** The process of using the measurement instruments to obtain measurements of the concept of interest.

**Interrogation:** The process of interrogating the validity of the systematized concept, the measurement instruments, and

their resulting measurements. Very often omitted or insufficiently rigorous in the typical ML approach to measurement.

**Lenses of validity:** Collectively used to interrogate the validity of the systematized concept, the measurement instruments, and their resulting measurements. Each lens—face validity, content validity, convergent validity, discriminant validity, predictive validity, hypothesis validity, and consequential validity—constitutes a different source of evidence about validity, as we explain in Appendix B.

**Conceptual debates:** Debates about precisely what is being measured and why—e.g., which meanings and understandings are reflected in the systematized concept? does it reflect the meanings and understandings that different stakeholders, all of whom may be interested in measuring the concept for different reasons, want it to reflect? if not, why? Conceptual debates encompass the systematization process and the parts of the interrogation process that directly focus on systematization and the systematized concept. As we note below, in some cases, the other parts of the interrogation process can also end up informing conceptual debates.

**Operational debates:** Debates about how the systematized concept is measured—e.g., do the measurement instruments yield valid measurements of the systematized concept? Operational debates encompass the operationalization process and the parts of the interrogation process that focus on operationalization, the measurement instruments, and their resulting measurements. In some cases, these parts of the interrogation process can also shed light on conceptual debates.

## B. Lenses of Validity

In this appendix, we explain the lenses of validity that we recommend using (Jacobs & Wallach, 2021), highlighting the roles each lens can play in conceptual and operational debates. We note that our views differ from those of Adcock & Collier (2001), who view the lenses of validity as only informing operational debates, treating conceptual debates as entirely separate. Instead, we argue that three lenses—face validity, content validity, and convergent validity—are especially useful for shedding light on conceptual debates. The other lenses can also reveal systematization issues even when used to interrogate the validity of the measurement instruments and their resulting measurements.

Each lens constitutes a different source of evidence about validity. We emphasize that determining what "sufficiently valid" means is one of the most difficult aspects of interrogating validity, as there are no universal or definitive answers. That said, the standard of evidence should be higher when the measurements are to be used for high-stakes purposes. Beyond this, as we discuss in Section 5, measurement instruments and measurements should always be accompanied by clear descriptions of the corresponding systematized con-

cepts, the various ways in which validity was interrogated, and the resulting evidence for—and against—their validity.

**Face validity:** Face validity focuses on the extent to which the systematized concept, in the case of conceptual debates, and the measurement instruments and their resulting measurements, in the case of operational debates, look reasonable. Face validity is therefore inherently subjective and must be supplemented with other, less subjective evidence. Face validity can be interrogated by anyone, including the people who systematized the concept, the people who developed the measurement instruments, the people who will use the resulting measurements, any other people who might be affected by the measurements, and any other stakeholders.

**Content validity:** In the case of conceptual debates, content validity refers to the extent to which the systematized concept reflects the most salient aspects of the background concept, while in the case of operational debates, content validity refers to the extent to which the measurement instruments align with the substance and structure of the systematized concept. Content validity therefore has two facets: *substantive validity* and *structural validity*.

In the case of conceptual debates, substantive validity focuses on whether the systematized concept fully specifies the observable phenomena that are connected to the concept. In the case of operational debates, substantive validity focuses on whether the measurement instruments align with those observable phenomena. In the case of conceptual debates, structural validity focuses on whether the systematized concept fully specifies the relationships between the observable phenomena and the concept and the relationships among the observable phenomena. In the case of operational debates, structural validity focuses on whether the measurement instruments align with those relationships.

As with face validity, content validity can be interrogated by anyone. However, because content validity has a much deeper focus than face validity, it is often best interrogated by stakeholders with specific expertise related to the concept of interest (in the case of conceptual debates) or the measurement instruments (in the case of operational debates). We note that when interrogating content validity involves familiarity with the specifics of particular GenAI systems or reviewing code, it can be difficult for stakeholders who lack sufficient technical expertise to be meaningfully involved.

**Convergent validity:** Convergent validity refers to the extent to which the measurement instruments yield measurements that are similar to measurements of the concept, or other similar concepts, obtained using other, already validated, measurement instruments. If the systematized concept is the same for both sets of measurement instruments, then convergent validity can be used to inform operational debates. If, however, the measurement instru-

ments use different systematized concepts (perhaps because they are intended to measure different, albeit similar, concepts) then it can be difficult to determine whether dissimilar measurements are due to systematization issues, operationalization issues, or both. As a result, convergent validity can inform both conceptual and operational debates.

**Discriminant validity:** Discriminant validity refers to the extent to which the measurement instruments yield measurements that are dissimilar to measurements of concepts that are dissimilar to the concept of interest (to the extent to which we believe they are dissimilar), obtained using already validated, instruments for measuring those concepts. Because dissimilar concepts must necessarily be systematized differently, it can be difficult to determine whether inappropriately similar measurements are due to systematization issues, operationalization issues, or both. Therefore, like convergent validity, discriminant validity can inform both conceptual and operational debates.

**Hypothesis validity:** Hypothesis validity focuses on the extent to which the measurements can be used to confirm hypotheses about the concept of interest that have already been confirmed via other methods. If the measurements do not confirm the hypotheses, this suggests systematization issues, operationalization issues, or both. To try to rule out systematization issues, it can be helpful to try to confirm the hypotheses using measurements obtained using other measurement instruments that operationalize the same systematized concept. If those measurements confirm the hypotheses, this suggests operationalization issues.

**Predictive validity:** Predictive validity focuses on the extent to which the measurements can be used to predict observable phenomena that are external to the concept of interest—i.e., distinct from those specified as part of the systematized concept—but known to be related to it in some way. Like hypothesis validity, if the measurements cannot successfully predict the external phenomena, this suggests systematization issues, operationalization issues, or both. Here too, it can therefore be helpful to try to predict the external phenomena using measurements obtained using other measurement instruments that operationalize the same systematized concept to try to rule out systematization issues.

**Consequential validity:** Consequential validity is concerned with the consequences of measurement,[10] including

---

[10]Consequential validity has a very different focus than the other lenses of validity. It was first proposed by Messick (1987), who argued that the consequences of measurement instruments and their resulting measurements should be fundamental to their validity. Consequential validity is also related to "Goodhart's Law" and "Campbell's Law" (Jacobs, 2021): Social scientists have long noted that the validity of measurement instruments and their resulting measurements can diminish over time as people focus on optimizing what is being measured, potentially also distorting the concepts of interest, the objects of interest, and even the contexts of interest,

1) the consequences of the systematization, operationalization, application, and interrogation processes and 2) the consequences of the systematized concept, the measurement instruments, and the measurements themselves. By focusing on the broader impacts of measurement—and especially the societal, ethical, and cultural impacts—consequential validity encompasses both intended and unintended consequences. This makes it the widest-ranging lens of validity.

## C. Framework Applicability

Below and in Figure 3, we illustrate the widespread applicability of the framework described in Section 3 by instantiating its levels with four very different measurement tasks: 1) measuring the prevalence of text that stereotypes social groups in the outputs of an already deployed LLM-based chatbot; 2) measuring the mathematical reasoning skills of a multimodal GenAI model; 3) measuring the extent to which a widely used LLM memorizes pieces of its training data; and 4) measuring the extent to which a company's GenAI assistant refuses to comply with harmful prompts.[11] We emphasize that these examples are not intended to be comprehensive. Rather, they are intended to illustrate how the framework applies to very different measurement tasks.

Suppose we wish to measure the prevalence of text that stereotypes social groups in the outputs of an already deployed LLM-based chatbot. The background concept encompasses all meanings and understandings of such text; the systematized concept might be specific definitions of "social groups" and "text that stereotypes social groups," as well as a particular set of linguistic patterns used by researchers in social psychology, sociolinguistics, anthropology, and other disciplines to define the presence of such text; the measurement instruments might be a judge LLM, a set of annotation guidelines that serve as prompt instructions for the judge LLM, and functions for aggregating the judge LLM's annotations; and the resulting measurement would then be the proportion of chatbot outputs that contain any of the linguistic patterns. We use this measurement task as a hypothetical running example throughout Section 4.

As another example, suppose we wish to measure the mathematical reasoning skills of a multimodal GenAI model (e.g., He et al., 2024). The background concept encompasses all meanings and understandings of mathematical reasoning skills; the systematized concept might be the accuracy of the model on highly challenging mathematical reasoning problems aimed at pre-university students, spanning algebra,

number theory, combinatorics, and geometry; the measurement instruments might be a particular set of International Math Olympiad problems, the corresponding scoring rubric, and a function for aggregating the per-problem scores into an overall accuracy—collectively comprising a benchmark; and the resulting measurement would then be the model's accuracy on the International Math Olympiad problems.

As a third example, suppose we wish to measure the extent to which a widely used LLM memorizes pieces of its training data. The background concept encompasses all meanings and understandings of memorization; the systematized concept might connect memorization to observable phenomena in the real world like regurgitation and extraction, justify the decision to focus on extraction, and provide specific definitions of extraction and its constituent concepts; the measurement instruments might be a particular set of pieces of training data, each split into a prefix (to be used as an input to the LLM) and a suffix (to be compared to the LLM's output for that prefix), a function for assessing whether a single LLM output contains exact or near-exact copies of a suffix, and an aggregation function for calculating the proportion of outputs that contain exact or near-exact copies of the suffixes; and the resulting measurement would then be the proportion of outputs that contain exact or near-exact copies of the suffixes when the LLM is prompted with the prefixes. We discuss this measurement task in Appendix E.

Finally, suppose we wish to measure the extent to which a company's GenAI assistant refuses to comply with harmful prompts. The background concept would be all meanings and understandings of refusal to comply with harmful prompts; the systematized concept might provide specific definitions of "refusal to comply" (e.g., specifying whether partial refusal is in scope, or only full refusal) and "harmful prompts" (e.g., "prompts where compliance would violate the company's terms of service"), as well as specifying observable phenomena in the real world that are connected to the concept of interest; the measurement instruments might be a particular set of harmful prompts (e.g., selected to span all relevant aspects of the company's terms of service), a judge LLM, a set of annotation guidelines that serve as prompt instructions for the judge LLM, and a function for aggregating the judge LLM's annotations into an attack success rate;[12] and the resulting measurement would then be the attack success rate for the GenAI assistant.

## D. Interrogation: Operationalization (Cont.)

In this appendix, we continue our discussion of interrogation from Section 4.2.2, focusing specifically on discriminant validity and hypothesis validity. Returning to the example

---

[11]We note that GenAI system inputs play different roles in these measurement tasks. This is because some measurement approaches treat inputs as measurement instruments (e.g., benchmarking and automated red teaming), while others (e.g., measuring the prevalence of some concept in real-world system outputs) do not.

[12]In keeping with the automated red-teaming literature, we follow the convention of using attack success rates to measure refusal, where a lower attack success rate means a higher refusal rate.

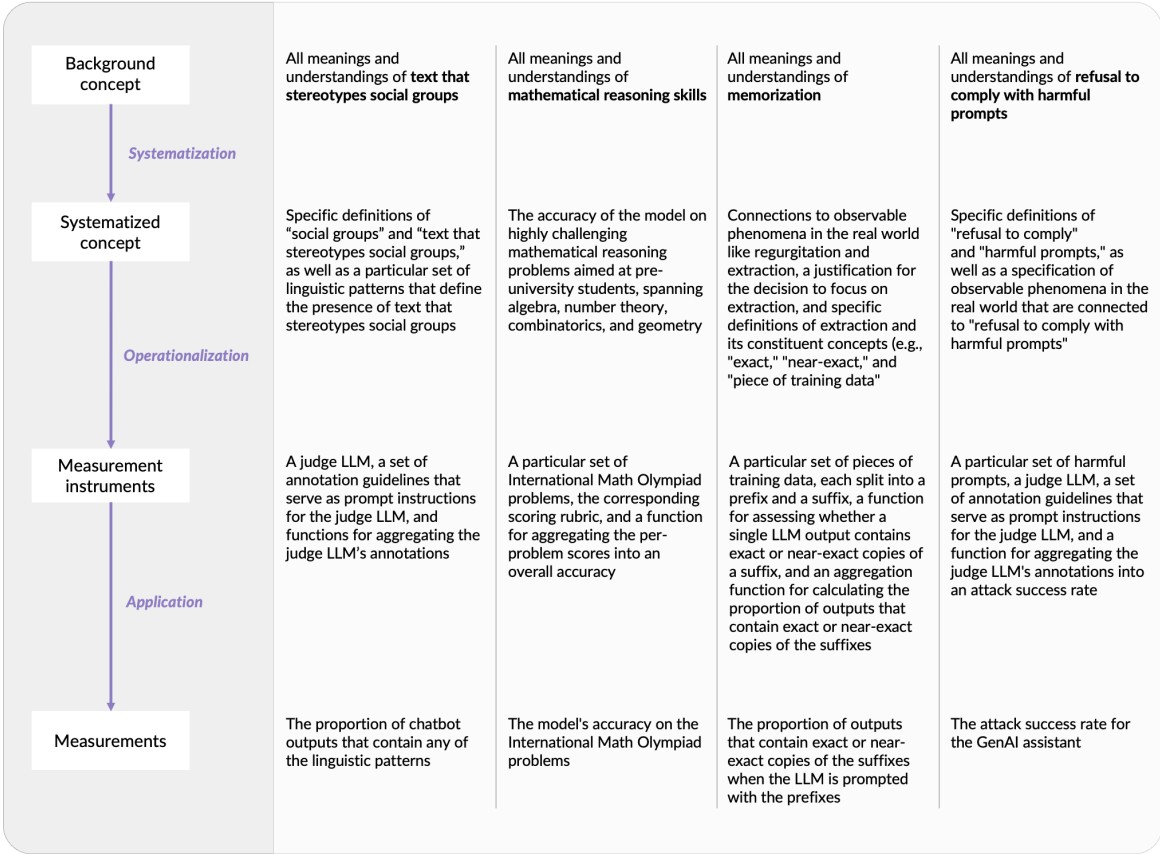

*Figure 3.* Instantiating the levels in the framework described in Section 3 with four very different measurement tasks.

of measuring the prevalence of text that stereotypes social groups in the outputs of an already deployed LLM-based chatbot, interrogating discriminant validity would involve comparing measurements obtained using our judge LLM to measurements of dissimilar concepts, such as hostile text or text with negative sentiment, obtained using other, already validated instruments for measuring those concepts. Although neither concept is completely unrelated to text that stereotypes social groups, it is important to check that our measurements only reflect them to the extent to which we believe they are similar to text that stereotypes social groups.

Hypothesis validity focuses on the extent to which the measurements can be used to confirm hypotheses about the concept of interest that have already been confirmed via other methods. For example, we might check that inputs designed to elicit text that stereotypes social groups (and confirmed via other methods) result in chatbot outputs that contain a higher prevalence of such text according to our judge LLM.

# E. Case Study: Memorization

In this appendix, we present a case study that shows how the framework described in Section 3 brings clarity to ongoing debates about the extent to which GenAI models encode exact or near-exact copies of pieces of their training data in their parameters—a concept known as memorization.

## E.1. Systematization

Memorization raises serious privacy and copyright concerns, and therefore lies at the center of several ongoing debates (Lee et al., 2025). However, it is an inherent attribute of a GenAI model that is very difficult to directly measure, especially without access to model internals. As a result, ML researchers and practitioners typically focus on measuring extraction (e.g., Carlini et al., 2023b; Nasr et al., 2025; Prashanth et al., 2025)—i.e., when a user intentionally and successfully prompts a GenAI model (sometimes as a component of a GenAI system) to generate such an output—or regurgitation (e.g., Aerni et al., 2025)—i.e., when a GenAI model (also sometimes as a component of a GenAI system) generates an output that contains an exact or near-exact copy of a piece of training data. Despite important differences between these concepts, they are often conflated, which has led to considerable conceptual confusion (Cooper & Grimmelmann, 2025).

Using the framework described in Section 3 would likely mitigate much of this confusion. As a background concept, memorization is inclusive of a broad range of meanings and understandings. For example, what counts as a "piece of training data?" A literal data point that was used during the training process? A portion of such a data point? Multiple such data points? As another example, how should "exact" and "near-exact" be defined? The answer to this question depends on the modality of the model outputs: For text, should we consider alternative spellings of a word? Slight changes in punctuation? What about translations? For images, should we focus on pixel differences? Semantic differences? Vector distances (Carlini et al., 2023a; Somepalli et al., 2023)? Providing specific definitions for these constituent concepts would help considerably, as would specifying how memorization is connected to observable phenomena in the real world like regurgitation and extraction and justifying the decision to focus on one over the other based on the purpose for which the measurements will be used. For example, if we are interested in making claims about typical use, then regurgitation is likely more appropriate. If we are instead interested in making claims about adversarial use, then extraction is likely more appropriate.

Ultimately, systematizing memorization involves many decisions, each influencing the resulting measurements. Explicitly foregrounding and interrogating these decisions would likely bring greater clarity to ongoing debates.

### E.1.1. INTERROGATION: SYSTEMATIZATION

Interrogating content validity makes it clear that although generating an output that contains an exact or near-exact copy of a piece of training data is strong evidence that the piece of training data was memorized by a GenAI model, any measurement of regurgitation or extraction is likely an underestimate of memorization (Lee et al., 2022; Carlini et al., 2021; Nasr et al., 2025; Carlini et al., 2023b). This is because the model may have memorized other pieces of training data that do not appear in its outputs. This then raises the question of what matters most to the evaluative claims we wish to make, touching on consequential validity. Do we only care whether a GenAI model generates exact or near-exact copies of pieces of training data? Or do we care whether it encodes exact or near-exact copies of pieces of its training data in its parameters, regardless of whether those pieces of training data are ever generated? These questions matter a great deal to ongoing privacy and copyright debates about memorization. For example, encoding exact or near-exact copies of pieces of training data in GenAI models' parameters is central to privacy and copyright debates about the nature of such models. For example, should GenAI models be considered personal or private data if they memorize such data (Nolte et al., 2025; Cooper et al., 2024)? Are GenAI models copies, in

a copyright-technical sense, of the pieces of training data they have memorized (Cooper & Grimmelmann, 2025)?

### E.2. Operationalization

Developing instruments for measuring memorization similarly involves many decisions. Focusing specifically on LLMs for concreteness, what if we do not have access to an LLM's training data? How might we develop a proxy for it? Exactly how many tokens best meets our definition of "a piece of training data"? 10? At least 32? 50? If we have chosen to focus on extraction, what prompting methodology should we use? One common approach, called discoverable extraction, involves sampling $2k$-token pieces from an LLM's training data (or a proxy for it). Each piece is split into a $k$-token prefix and a $k$-token suffix. The prefixes are used as inputs to the LLM, while each suffix is compared to the LLM's output for the corresponding prefix, usually obtained using deterministic, greedy sampling (Carlini et al., 2021; 2023b; Nasr et al., 2025). This approach is relatively inexpensive, but its use of deterministic, greedy sampling, rather than non-deterministic, non-greedy sampling, which is more common when using LLMs in practice, may result in unrealistic measurements. Another approach takes a probabilistic perspective, sampling each prefix multiple times to approximate the probability of generating that prefix's suffix using non-deterministic, non-greedy sampling (Hayes et al., 2025). For either approach, which pieces of training data should we use? Which function best meets our definition of "exact" and "near-exact"? Each of these decisions influences the resulting measurements—often significantly—so foregrounding and interrogating them is especially important.

### E.2.1. INTERROGATION: OPERATIONALIZATION

We emphasize that several key findings from prior work on memorization can be reinterpreted using the lenses of validity. First, again focusing specifically on LLMs for concreteness, when deciding how many tokens of training data to use, it is important to use enough tokens to ensure that the resulting measurements genuinely reflect pieces of training data that have been encoded in an LLM's parameters, as opposed to happenstance generation (Carlini et al., 2021; Nasr et al., 2025). Interrogating face validity has led to the general consensus that 10 tokens is too short to confidently rule out happenstance, with 50 tokens now accepted as the norm (Nasr et al., 2025). Similarly, many pieces of data (regardless of whether they are pieces of training data) could, theoretically, be generated by happenstance, given enough attempts. Recent work on probabilistic discoverable extraction has therefore interrogated discriminant validity by comparing measurements of extraction to measurements of the rate of generating exact or near-exact copies of pieces of unseen test data that, by definition, could not have been memorized (and therefore cannot be extracted) (Hayes et al.,

2025). Finally, also focusing on probabilistic discoverable extraction, recent work has interrogated convergent validity by assessing whether measurements of extraction are correlated with the corresponding suffixes' perplexities, as expected, finding that they are (Hayes et al., 2025).

