# OpenReview forum: "Position: Evaluating Generative AI Systems Is a Social Science Measurement Challenge"
_ICML.cc/2025/Position_Paper_Track — ICML 2025 Position Paper Track poster_

### Official Review · Reviewer_xuVN · 2025-02-23

**Significance:** 2
**Argument Clarity:** 3
**Rating:** 3
**Confidence:** 4

**Questions:**

- Same question as stated in the first bullet under the weakness section.

**Discussion Potential:**

2

**Paper Summary:**

The paper proposes a four-level framework to evaluate Gen AI system performance that is rooted in social science that involves defining background concept, systematized concept, measurement instruments, and measurements. Authors suggest utilizing interrogation techniques such as face validity, content validity, convergent validity, discriminant validity, predictive validity, hypothesis validity, and consequential validity from measurement theory to examine the validity of measurement instruments and measurements and argue that interrogating validity without considering context is not possible.
Authors discuss Gen AI performance evaluation in the context of measuring the stereotyping behaviors of LLM-based systems, mathematical reasoning skills of a GenAI system, and measuring the extent to which a GenAI system regurgitates to show the shortcomings of typical AI evaluation approaches.

**Position:**

Yes

**Position In Title:**

Yes

**Related Work:**

3

**Strengths And Weaknesses:**

**Strengths**
- The idea of using social science
- The paper's use of a common theme example (i.e. demeaning text in the outputs of an LLM-based system) to clarify the discussions is very helpful and helps reader to get a better grasp of some of the abstract topics discussed in the paper.

**Weaknesses**
- The paper arguments about paying more attention to systematization and operationalization makes sense. However, the paper does not provide clear evidence that by using the proposed framework we can achieve more accurate measurements. For example, in the case of "stereotyping behaviors of LLM-based systems", are there existing studies that demonstrate misleading or inaccurate measurements from StereoSet and CrowS-Pairs that could have been avoided by utilizing social science principles?
- It would have been more compelling if the ideas in this paper were applied to a concrete domain to achieve two goals: 1- highlight the shortcoming of existing approaches; 2- providing concrete guidelines of how the underlying principles of the proposed framework could be implemented in real world.

**Support:**

2

---

> ### Author Rebuttal · Authors · 2025-04-01
>
> We thank the reviewer for their thoughtful comments.
>
> ### W1: Contribution of the proposed framework
> Our position paper offers conceptual resources to support researchers and practitioners in conducting and critically interrogating AI evaluations. The measurement framework we offer here is not (just) about “more accurate” measurements.  Indeed, an important implication of our arguments is that measurement “accuracy” isn’t even well-defined absent precise systematization (see response to Reviewer igKg).   We believe that by standardizing around the measurement framework widely used within the social sciences, and associated terminology, we can make significant collective progress towards maturing current evaluation practices into a “science of evaluations.”  Our work is in the category of a “call to action” and “recommendation for changes to how we conduct and evaluate research.” A complete end-to-end instantiation of our framework is a full research contribution in and of itself, so to make our arguments we synthesize from the ample evidence that already appears, albeit very piecemeal, in existing literature.
>
> ### W2: Application of framework
> While a complete end-to-end instantiation of our framework would be outside the scope of a position paper, we propose to add (either to the main text, space permitting, or to the appendix) two short (about 1-2 page) discussions, one situating recent work on LLMs-as-judges in the context of our framework (see also Reviewer igKg Q1 response) and one on measurement and memorization of training data (currently discussed briefly in Section 3). We will also strengthen our discussion of existing work in RAI (e.g., lines 377-8 where we discuss the Blodgett et al 2021 critique of popular stereotype benchmarks).
>
> For the discussion of measurement and memorization, we propose a 1-2 page meta-study of the most common way to quantify memorization, “discoverable extraction” [1,2,3], which is used widely in research and model-release reports [e.g., 4,5,6]). Recent work [7] shows how discoverable extraction is, in our parlance, not well systematized. Instead there has been a leap directly to operationalization—measuring extraction in one shot with deterministic greedy sampling. Upon further scrutiny, this operationalization seems to lack content validity and reliability, particularly by failing to account for non-determinism in typical LLM interaction patterns. Instead, [7] takes a step back to develop a new systematization that incorporates the non-deterministic nature of LLMs. Instead of centering on deterministic, one-shot, greedy sampling, their systematization allows for multiple queries to the model to compute the probability of extracting a given piece of training data. This conceptualizes extraction as a stochastic process, and accommodates cases of extraction that deterministic, greedy sampling misses. The authors test several operationalizations of that systematization. Altogether, the authors obtain measurements of memorization that are more reliable and also lend themselves to more valid comparisons of extraction rates across models. For example, since their probabilistic measure catches cases of extraction that greedy sampling misses, their measurements show that discoverable extraction significantly underestimates total possible extraction. The discussion that we propose to add will include more depth about the background concept, systematized concept (including conceptual debates in the systematization process), measurement instruments (including operational debates and the operationalization process), and measurements.
>
> [1] Carlini et al. 2021. Extracting training data from large language models.
>
> [2] Carlini et al. 2022. Quantifying memorization across neural language models.
>
> [3] Nasr et al. 2023. Scalable extraction of training data from (production) language models.
>
> [4] Gemini Team. 2024. Gemini 1.5: Unlocking multimodal understanding across millions of tokens of context.
>
> [5] Biderman et al. 2023. Pythia: A suite for analyzing large language models across training and scaling.
>
> [6] Llama Team. 2024. The Llama 3 Herd of Models.
>
> [7] Hayes et al. 2024. Measuring memorization in language models via probabilistic extraction.

---

> > ### Comment · Reviewer_xuVN · 2025-04-05
> >
> > I'd like to thank the authors for their responses and further explanations. In light of authors' explanations for "Application of framework" I changed my overall recommendation to "weak accept".

---

### Official Review · Reviewer_N8WB · 2025-03-09

**Significance:** 3
**Argument Clarity:** 3
**Rating:** 4
**Confidence:** 3

**Questions:**

Many of my concerns/questions are already addressed in the strengths and weaknesses section. If the authors can provide convincing responses to these issues or revise the paper accordingly, I would be happy to improving my evaluation score.

**Discussion Potential:**

3

**Paper Summary:**

The position paper argues that evaluating generative AI systems is fundamentally a social science measurement challenge. It contends that current evaluation practices in ML are often inconsistent and insufficient because they fail to capture abstract and contested concepts with the necessary rigor. The authors propose adopting a four-level framework, adapted from Adcock & Collier (2001), to structure the evaluation process. By separating conceptualization from operationalization, the paper emphasizes the need for interdisciplinary collaboration and multiple lenses of validity to ensure robust and meaningful evaluation of GenAI systems.

**Position:**

Yes

**Position In Title:**

Yes

**Related Work:**

3

**Strengths And Weaknesses:**

Strengths:
1. This work tackles a pressing issue for the ML community – how to evaluate complex generative models reliably. As GenAI systems become pervasive, their evaluation (for safety, fairness, capability, etc.) has emerged as a critical bottleneck. The paper’s call to improve and standardize evaluation is relevant to ICML researchers and practitioners who are striving to compare models and ensure their systems are trustworthy.
2. The paper provides an interdisciplinary perspective on ML evaluation. It effectively reframes the evaluation of AI models (especially generative models) as akin to measuring latent constructs in social sciences.
3. A strong aspect of the paper is that it is grounded in established measurement theory and related work. The authors explicitly cite seminal works (e.g. Adcock & Collier 2001 for the multi-level measurement framework, Cronbach & Meehl 1955 and Messick 1996 on validity theory) to show the provenance of their ideas.

Weaknesses:
1. While the paper presents a compelling framework, the practical implementation of the four-level framework may need deeper development or clarification. For instance, the paper emphasizes the need for an “explicitly systematized concept” before designing an evaluation, but it provides relatively limited guidance on how researchers should negotiate and pin down that concept in practice (especially when faced with disagreements or ambiguity). The paper could more robustly address how to carry out each stage of the framework, perhaps by elaborating one of the examples step-by-step. Without such a demonstration, some readers might find the proposal a bit high-level – intuitively appealing, but not fully actionable.
2. As a position paper, the contribution mainly lies in synthesizing ideas from social science and applying them to ML, rather than introducing brand-new empirical findings or algorithms. While this interdisciplinary synthesis is valuable, one could argue that many of the individual components of the argument have been voiced before in various forms. For example, the importance of construct validity and the pitfalls of common ML benchmarks have been discussed in the ML ethics and NLP communities.
3. One concern is whether ML researchers will find the framework practical and tailored to AI. The social science paradigm typically involves measuring human attributes, and some ML practitioners might be skeptical of directly applying those methods to AI systems. The authors do clarify in their impact statement that they don’t mean to naively reuse human assessment tools for AI, but this nuance appears outside the main text.
4. The paper is generally well-referenced, but there are a few areas where additional citations or context could strengthen its claims. For example, when discussing the need for standardized evaluation, the authors cite several works and the AI Index report, but they might also mention efforts like Holistic Evaluation of Language Models (HELM) or other benchmark consolidation projects in NLP, which attempt to catalog many metrics. Also, there are other competing proposal's on advancing measurement theory in AI research by drawing inspiration from social sciences, such as Morehouse et al. (2025).

Reference:

[1] Liang, P., Bommasani, R., Lee, T., Tsipras, D., Soylu, D., Yasunaga, M., Zhang, Y., Narayanan, D., Wu, Y., Kumar, A., Newman, B., Yuan, B., Yan, B., Zhang, C., Cosgrove, C., Manning, C. D., & Ré, C. (2022). Holistic evaluation of language models [Preprint]. arXiv. https://doi.org/10.48550/arXiv.2211.09110

[2] Morehouse, K. N., Swaroop, S., & Pan, W. (2025). Rethinking LLM bias probing using lessons from the social sciences. arXiv preprint arXiv:2503.00093. https://doi.org/10.48550/arXiv.2503.00093

**Support:**

3

---

> ### Author Rebuttal · Authors · 2025-04-01
>
> Thank you to the reviewer for their thoughtful response. We are happy to see that the impact of this work to intervene on how GenAI evaluation is done is clear. That said, in light of the expressed concerns, we will emphasize the applicability of our framework in the text.
>
> ### W1: Practical implementation of the framework
> Thank you for raising this point. We will show in the text that this framework is directly actionable. Specifically, we will emphasize that the first priority of this framework is to separate systematization from operationalization, and the second is to interrogate the validity of measurement instruments and their resulting measurements. The latter is hard to do well (i.e., meaningful and reliable AI evaluation is hard to do well) without systematizing the background concept.
>
> We will more clearly connect these priorities to our examples in Section 3, and will highlight this applicability throughout the text. Different systematizations of mathematical reasoning, for instance, clearly require different measurement instruments (PhD level proofs? Basic financial reasoning tasks? The SAT?) and will produce different outputs.  The memorization example in that section also reveals where there might be disagreement (adversarial attacks? Length of regurgitated materials?), where negotiating that disagreement will depend on the goals of the researcher.
>
> (Please see also our response to Reviewer igKg, W1 and Reviewer xuVN, W2.)
>
> ### W2: Value of interdisciplinary synthesis
> While some of these ideas have been introduced before, their uptake by the GenAI community has not been very successful nor systematic. (See also our response to Reviewer igKg, W2.) Researchers and practitioners are still struggling to meaningfully draw on and incorporate these ideas into evaluation frameworks. By offering a cohesive framework that synthesizes and reinterprets ideas to change what it means to do meaningful evaluation, we aim to provide a path for better uptake in the ML and NLP communities more broadly.
>
> ### W3: Interpretation of social science measurement framework by ML audience
> Thank you for highlighting this point, we agree that some ML researchers may misinterpret us as suggesting that they adopt existing measurement instruments from the social sciences. As you noted, we directly address this potential misunderstanding in our impact statement, where we clarify the distinction between the measurement framework from the social sciences and specific measurement instruments from the social sciences. There, we emphasize that rather than human instruments in the social sciences being equivalent to AI evaluations, instead, the measurement framework offers a principled way to develop and interrogate AI evaluations.
>
> We will move the relevant text from the impact statement to the main text, especially to clarify that we are not advocating for treating AI evaluations as equivalent to human assessment tools.
>
> ### W4: Clarify connections to existing evaluation frameworks
> We will add citations to HELM and other benchmark consolidation projects that catalog many metrics in our revision (BIG-bench,  DecodingTrust, TrustLLM). In doing so, we will clarify how such standardized evaluation efforts would benefit from engaging with our framework:. Though these benchmark suites provide standardized conditions (i.e., datasets, model hyperparameters) for evaluating and comparing models, the benchmarks in these suites do not share a standardized measurement process. We note that the HELM authors recognize this weakness but leave systematization and validation to future work: for systematization, see e.g., p. 32: “We recognize that this work fails to articulate a more precise (operationalized) definition of toxicity.... we do believe there is ample room for improvement...” and for validation, see, e.g.,  p. 82: “Consequently, the quality and useful of our benchmark is contingent on this assumption: we encourage future work to interrogate the validity of our datasets and to introduce protocols to help ensure the validity of future datasets.”
> Morehouse et al. (posted on arxiv after the ICML submission deadline) is a narrower approach than we provide, focusing on a stylized task to interrogate LLMs focusing on ecological validity and bias, while we encompass all aspects of validity and measurement of topics beyond bias. We will update our text to add a reference and show how the work is/n’t related.
>
> In our discussion of the need for standardized evaluation, we will also add a citation to BetterBench, a quantitative evaluation of the usability and design of benchmarks.   Our framework complements such efforts: where BetterBench can help surface specific weaknesses in benchmark design for practitioners, our paper provides the conceptual framework needed to thoroughly interrogate and improve evaluations.

---

> > ### Comment · Reviewer_N8WB · 2025-04-03
> >
> > Thanks for the response. The authors have addressed most of the questions I raised. Given this, I have updated my score accordingly.

---

### Official Review · Reviewer_igKg · 2025-03-17

**Significance:** 3
**Argument Clarity:** 3
**Rating:** 3
**Confidence:** 3

**Questions:**

- In today's eval landscape, especially in the context of LLMs, there is an increasing adoption of LLM-as-judges, which allows for automated evals of free form outputs. However, in the context of your framework, LLM-as-judge muddies the line between an indicator and a measurement tool. For example in your running example of hateful content generated by an LLM, a judge based eval would take in a prompt that is defines your indicator and hope that the judge "understands" the prompt and reliably measures the presence of an indicator. Would this not make it even harder for stakeholders to be involved in having a say in making a benchmark? ML researchers barely understand how prompting truly works, and if the measured quantity does not satisfy face validity, how would an affected stakeholder be able to interrogate the systematization and/or operationalization process? Would your framework be at odds with auto-evals like LLM-as-a-judge?

 - If I were running a small lab trying to build open source GenAI systems to democratize research on these systems; I would likely not have the resources to work through a framework like yours to build proper evals (even if I would ideally like to). However, big industry labs, which often keep many details including even weights of these models secret, will likely have the resources to invest in properly constructing evals that engage multiple stakeholders, interrogate the systemization and operationalization of a concept and refine their benchmarks as needed. Do you think there could be a relaxation of your framework to get somewhat noisy eval scores without having to go through the entire process?

**Discussion Potential:**

3

**Paper Summary:**

The paper argues that the current paradigm of evals in GenAI (and ML more generally) could benefit by drawing upon methods from measurement theory, an approach often used in social sciences to measure abstract concepts. Such an approach disentangles the concept being measured from the methodology used to measure these concepts and thus allows for a more granular introspection of an evaluation pipeline. The paper also argues that such a distinction between systemization and operationalization enables various stakeholders to participate in the process of evaluating the very systems that they will interact with. More concretely, the paper proposes a framework that clearly differentiates between systemization of a concept (i.e., going from an abstract concept to defining concretely what needs to be measured), operationalization of the defined concept (which involves implementing exactly how the well defined concept will be measured), and finally applying this implementation to obtain measurements of the concept. Such a framework allows interrogating and analyzing many facets of validity of the measured evals of a system.

**Position:**

Yes

**Position In Title:**

Yes

**Related Work:**

2

**Strengths And Weaknesses:**

Strengths:

1. The paper takes a clear stance about evaluating GenAI systems and how the current state of evals is rather messy and can benefit by borrowing from measurement in social sciences.

2. There are many interesting perspectives provided in the paper about how current evals / benchmarks fall short of truly engaging with what they are exactly trying to measure, leading to misleading and non-actionable benchmarks.

3. I also liked how the paper argues that by cleanly separating systemization and operationalization one can actually interrogate which part of the pipeline needs to be re-examined in order to fix the validity of measurements. Such a separation also allows to iterate on the measurement process which is very necessary in order to build meaningful evals.

Weaknesses:

1. While I like the stance in the paper, it operates mostly at a rather abstract level; I know part of this is intentional, since the paper is proposing a framework, not a concrete benchmark. However, I would like the paper to engage a bit more with how such a framework relates to the realities of building and evaluating GenAI systems today. Should such a thorough eval be done before deployment to end users? Before releasing model weights publicly on a platform? The promise of these systems (over traditional ML systems) is that they can solve a vast breadth of tasks "zero shot"; thus loads of benchmarks (MMLU etc) today focus on breadth, instead of depth; and maybe that's fine if a model is being released for research purposes. Perhaps a framework like yours is necessary when, eg, finetuning a model that's decent on these coarse, ill defined evals for actual deployment with end users. Some discussion around the realities of how GenAI models are built today vs how your framework can be incorporated into the building process would really strengthen the paper.

2. Perhaps related to #1, I would have liked to see more focus on why this framework applies more to GenAI systems than traditional ML systems (the paper does mention one could / should apply this to also traditional ML systems). Is it because GenAI systems are more mainstream? Or is it because GenAI, by design, has the ability to give free form output that will inherently have abstract concepts? I would've liked to see how such a framework needs to be adapted for traditional ML vs GenAI systems.

3. The paper seems to suggest that ML literature has not dealt with measuring abstract concepts; however, this is not true if you look at the literature on responsible AI. For example a lot of work on fair ML actually explicitly differentiates the concept being measured and the operationalization of the measurement. I would have liked to see some discussion on how many works in fairness/explainability of traditional ML systems (that actually did borrow many tools from social sciences) can be extended now to GenAI systems.

**Support:**

2

---

> ### Author Rebuttal · Authors · 2025-04-01
>
> Thank you for your thoughtful review. In our updated draft, we will more clearly indicate how our framework relates to the realities of building and evaluating GenAI systems.
>
> ### W1: Connecting abstract prescriptions to GenAI today
> Our framework is relevant for any type of benchmark (broad or highly specialized) or evaluation, and can therefore inform the meaningful development and deployment of models and systems at multiple stages of the GenAI pipeline. By emphasizing and separating systematization and operationalization, it is possible to head off many issues with existing benchmarks and evaluations by design. For instance, recent work (https://arxiv.org/pdf/2502.06559) raises issues, including validity, with current benchmarks that would be less likely to arise if their design was informed by the framework we propose. Our framework can also be used to reveal how benchmarks and evaluations may not actually capture what they purport to. We will add these points to the paper.
>
> See also response to reviewer N8WB, W1.
>
> ### W2: Relevance for GenAI vs. traditional ML
> Our framework applies to both traditional ML as well as GenAI. However, GenAI systems are open-ended (in terms of inputs, outputs, and use cases, purported to be “general purpose” but with a wide range of risks to society. This open-endedness means that the space of challenges and contestations in these measurement processes are more complicated; so while relevant to all ML systems, especially those that are widely deployed, the consequences of not using this approach are arguably more severe for GenAI. We will highlight this point more clearly.
>
> ### W3: Connections to Responsible AI literature
> Thank you for this point; the RAI community has engaged with social science and measurement and benefited greatly when it has done so. (A notable example is Savoldi et al 2021 https://tinyurl.com/3vtpuv8a who use sociolinguistics to conceptualize bias.) Despite this, it has been repeatedly demonstrated that measurements of “bias” in generative language and image models have often skipped systematization, resulting in assessments that measure very different things and are challenging to interpret (Blodgett et al 2020; Goldfarb-Tarrant et al 2023 https://tinyurl.com/4z4w23cw). This motivates our work: even in the RAI community, researchers and practitioners are still struggling to meaningfully draw on and incorporate these ideas into evaluation frameworks, and their uptake by the broader GenAI community has not been very successful nor systematic. By offering a cohesive framework, we aim to provide a path for better uptake, in turn helping the GenAI community (including the RAI community) avoid common shortcomings identified in the literature.
>
> ### Q1: LLM-as-a-judge
> LLM-as-judge is a great example of a setting where our framework can be used to clarify distinctions that are muddled in existing literature.  There are a number of ways to operationalize measurement instruments: LLM-as-a-judge, manual human annotation, or even string-matching functions (lines 15-45 in https://tinyurl.com/mpdm6wsb show prefixes used to define jailbreaks). Now consider the demeaning example: carefully systematizing demeaning enables more specific annotation guidelines—and enables us to set up the validation task to evaluate the accuracy of a given operationalization of the concept (LLM-as-judge, simple non-LLM classifier, majority vote over K human annotators, etc). For an output like “his ability to code despite being blind is inspiring”, the LLM-as-judge might say this is not demeaning. Is that correct? Under a systematization that includes “inspiration porn” as a demeaning language pattern, this text is demeaning; the judge is wrong. By having stakeholders weigh in at the systematization stage, we make such decisions explicit, enabling coherent validations of LLM-as-judge performance.
>
> We propose to replace our in-text discussion of human annotation in section 2.2 with LLMs-as judges to highlight how our framework is relevant to popular practices.
>
> ### Q2: Practical realities of a small lab
> The most important contribution of this work is our call to distinguish systematization vs. operationalization and to validate systematizations (to whatever depth is desired). Despite the size and access to resources in large labs, systemization and operationalization are still almost always conflated in academia and industry. Indeed, small labs can use this framework to disrupt the sloppy efforts of large organizations, using different types of validity to interrogate or improve models strategically.

---

### Decision · Program_Chairs · 2025-04-29

**Decision:**

Accept (poster)

**Comment:**

The position paper tackles a very important and timely problem of properly evaluating AI systems. The paper convincingly argues that ideas from social science measurement approaches are a promising way forward and propose a high-level framework for doing so. While some of the ideas have been 'floating around piecemeal', the paper manages to bring everything together in a cohesive framework.
The initial concerns of the reviewers circled around connections to other evaluation frameworks and doubts whether the proposed framework is practically feasible and beneficial. The authors managed to address all major concerns of the reviewers in the rebuttal, with some promised additional clarifications and a significantly extended discussion on the applicability of the framework and how it complements other approaches and initiatives. The reviewers confirmed that these additions nicely address their concerns by grounding the ideas in the position paper (rather than proposing a concrete a novel technical contribution/framework solving the identified problem which would be out of scope for a position paper).